# Preparation, Drug Distribution, and In Vivo Evaluation of the Safety of Protein Corona Liposomes for Liraglutide Delivery

**DOI:** 10.3390/nano13030540

**Published:** 2023-01-29

**Authors:** Ruihuan Ding, Zhenyu Zhao, Jibiao He, Yuping Tao, Houqian Zhang, Ranran Yuan, Kaoxiang Sun, Yanan Shi

**Affiliations:** 1School of Pharmacy, Key Laboratory of Molecular Pharmacology and Drug Evaluation, Ministry of Education, Collaborative Innovation Center of Advanced Drug Delivery System and Biotech Drugs in Universities of Shandong, Yantai University, Yantai 261400, China; 2School of Life Science, Yantai University, Yantai 261400, China

**Keywords:** liposome, oral drug delivery systems, osmoregulatory peptide, mucus layers, intestinal epithelial permeability

## Abstract

The development of oral drug delivery systems is challenging, and issues related to the mucus layer and low intestinal epithelial permeability have not yet been surmounted. The purpose of this study was to develop a promising formulation that is more adapted to in vivo absorption and to facilitate the administration of oral liraglutide. Cationic liposomes (CLs) linked to AT-1002 were prepared using a double-emulsion method, and BSA was adsorbed on the surface of the AT-CLs, resulting in protein corona cationic liposomes with AT-1002 (Pc-AT-CLs). The preparation method was determined by investigating various process parameters. The particle size, potential, and encapsulation efficiency (EE%) of the Pc-AT-CLs were 202.9 ± 12.4 nm, 1.76 ± 4.87 mV, and 84.63 ± 5.05%, respectively. The transmission electron microscopy (TEM) imaging revealed a nearly spherical structure of the Pc-AT-CLs, with a recognizable coating. The circular dichroism experiments confirmed that the complex preparation process did not affect the secondary structure of liraglutide. With the addition of BSA and AT-1002, the mucosal accumulation of the Pc-AT-CLs was nearly two times lower than that of the AT-CLs, and the degree of enteric metaplasia was 1.35 times higher than that of the PcCLs. The duration of the intestinal absorption of the Pc-AT-CLs was longer, offering remarkable biological safety.

## 1. Introduction

Protein and peptide drugs possess both chemical and protein properties and offer the advantages of high activity, safety, specificity, selectivity, and good drug-formation capability. Due to patient compliance and ease, oral administration is the most preferred route. However, due to the key physical and chemical properties of proteins and peptides, including their vulnerability to enzyme degradation and poor permeability through the intestinal mucosa, their absorption is incomplete [1]. Due to their poor oral bioavailability, many drugs must be injected intravenously rather than administered orally, rendering such treatments expensive [2]. Currently, polypeptide drugs are widely used in the treatment of many diseases, including diabetes, which is a metabolic disease. Diabetes is mainly caused by insufficient insulin secretion and is mainly treated by insulin or glucagon-like peptide-1 (GLP-1) analogs and other drugs. Among these drugs, liraglutide has 97% structural homology to endogenous human GLP-1 and can stably resist metabolic degradation by dipeptidyl peptidase-4 (DPP-4). Its half-life after subcutaneous administration is 13 h, implying that it needs to be administered once daily [3,4]. Its most common adverse reactions include gastrointestinal problems, which range from mild to moderate. Treatment costs are high. Without insurance, the average out-of-pocket expenses per month exceed USD 1000. Moreover, liraglutide can only be administered by subcutaneous injection, which may pose difficulties for some patients [5]. Thus, it is necessary to develop an oral preparation of liraglutide. The improvement in oral bioavailability is a major challenge faced by formulators in the development of successful products [6]. This endeavor requires a deep understanding of the behavior of an oral drug delivery system in the gastrointestinal tract during the development of the preparation process, and it is recommended that this preparatory process must be conducted according to the drug’s nature; indeed, it is important to predict the drug’s in vivo behavior [7]. The pressing questions concerning the development of oral drug delivery systems are as follows: (1) How can drug design be guided by the information obtained from oral delivery barrier characteristics? (2) What are the stability-related issues of peptide drugs before they arrive at the site of action? (3) Which structural features of nano-formulations affect their intestinal absorption efficiency?

Concerning oral drug delivery systems, the mucus layer is the first barrier to the passage of exogenous substances. Mucus is produced by specialized cells and secreted apically as a protective and lubricating fluid [8]. The second barrier is the phospholipid bilayer of the epithelial cell itself and the intracellular environment [9]. Nanoparticles initially penetrate through the mucus layer to gain entry to the surface of epithelial cells, and most nanocarrier transport relies on endocytosis (the involvement of endosomal capture can lead to inefficient delivery) [10,11]. Additionally, nanoparticles can also enter cells through the paracellular route [12]. The paracellular pathway is composed of tight junctions (TJs) maintained by a complex network of protein interactions. TJs are selectively permeable barriers that often represent a rate-limiting step in paracellular trafficking [13]. However, a TJ chain is a dynamic structure, and nanocarriers can temporarily and reversibly open intercellular TJs by using auxiliary agents or smart formulations to penetrate cells [14]. Existing transcellular penetration enhancers (PEs) acting on the paracellular pathway have been shown to enhance the permeability of macromolecules [15]. For example, researchers have constructed citric acid cross-linked shells made of carboxymethylcellulose wrapped around core nanoparticles for efficient transcellular transport by reversibly opening tightly connected paracellular transport pathways [16]. Anthocyanins from strawberries can enhance the intestinal permeability of drugs, as evidenced by the screening of polyphenolic compounds isolated from plants [17].

AT-1002 is a hexamer synthetic peptide (FCIGRL) and is the active domain of the second toxin of Vibrio cholerae, namely, the zonal closure toxin (Zot). Zot can bind to putative surface receptors, activate intracellular signaling, and, finally, decompose TJs. Due to the expression and purity of prokaryotic toxins, some side effects can occur; thus, synthetic peptide AT-1002 was developed to enhance intestinal permeability and avoid the abovementioned problems [18]. AT-1002 can cause the TJ opening of the Caco-2 cell monolayer [19,20], and this effect is reversible. The mechanism of action of AT-1002 can be reversed by octapeptide azapeptide [21], which is a reversible TJ regulator that functions via the inhibition of zonlin [22]. In our previous study, chitosan (CS) nanoparticles linked to AT-1002 as a core and poly-N-(2-hydroxypropyl) methacrylamide (pHPMA) as a smart escape in a core–shell structure overcame the pH and mucus barriers, demonstrating that this delivery system significantly enhanced the oral hypoglycemic effect of liraglutide [23].

A liposome is a self-assembled spherical vesicle system containing water-borne nuclei, showing the characteristics of biodegradability and high biocompatibility. The physical and chemical properties of liposomes can be changed, including particle size, surface charge, lipid composition, and bilayer fluidity, and can be used to design effective drug carriers for diseases [24]. In recent years, cationic liposomes (CLs) have been used as oral drug carriers, owing to their increased permeability through intestinal cells via their electrostatic interactions with the intestinal epithelial cell membrane. Therefore, CLs were selected as the oral drug delivery agents in this study. Cationic amphiphilic compounds are the main components of CLs, which are usually used for their positively charged lipids and polymers. The use of polymer carriers to modify the surface of liposomes to design more effective drug delivery systems is an effective preparation method [25].

Bovine serum albumin (BSA) is a plasma protein widely distributed in bovine blood, with a relative molecular weight and an isoelectric point of 66.5 kDa and 4.7, respectively. As endogenous macromolecules from a wide range of sources, these plasma proteins are widely used in the preparation of albumin-derived prodrugs and albumin-loaded nano-carriers [26]. BSA, as a part of the mucus-penetrating carrier, is degraded by trypsin in the process of penetrating the mucus, thereby exposing the core structure for the oral delivery of insulin [27]. CLs can be prepared by adding cationic materials to the nano-prescription of liposomes. Because albumin is negatively charged at a neutral pH, it can be adsorbed on the surface of the cationic nanocarrier through electrostatic attraction, changing the positive charge surface of the cationic nanocarrier and resulting in electrically neutral and hydrophilic nanoparticles.

There are few related studies for improving the oral delivery of peptides that involve both protein-crowned liposomes combined with cellular bypass-regulating peptides. Therefore, in this study, we used BSA as a protein coating with CLs attached to an osmoregulatory peptide as the core to improve mucus permeability and enhance absorption efficiency in the intestine. First, the physicochemical properties, encapsulation efficiency, drug release, and stability of the optimal formulation of the liraglutide protein corona CLs were evaluated. Through the mucus and intestinal uptake experiments, the protein coating was found to significantly improve intestinal absorption efficiency. The IVIS spectral imaging system and hematoxylin–eosin (HE) staining were used to evaluate the distribution and biological safety of the liraglutide protein corona cationic liposomes in vivo.

## 2. Materials and Methods

### 2.1. Materials

Distearoyl phosphatidylcholine (DSPC) was purchased from Penske Biologicals (Shanghai, China). Chol-PEG-AT-1002 and FITC-liraglutide were purchased from the Shanghai Qiang Yao Biochemical Co., Ltd. Cholesterol (Chol) was purchased from the RVT Pharmaceutical Technology Co., Ltd. (Shanghai, China). Fluorescein isothiocyanate (FITC) and mucin were purchased from the Yuanye Biochemical Co., Ltd. (Shanghai, China). N-acetyl-l-cysteine (NAC) was procured from the Shanghai Macklin Biochemical Co., Ltd. (Shanghai, China). Mice were purchased from the Gem Pharmatech Co., Ltd. (Jiangsu, China).

### 2.2. Prescription Screening

DSPC is a synthetic phospholipid that is widely used in pharmaceutical preparations. In clinical liposome products, Onivyde (irinotecan liposome injection) [28] and DaunoXome (daunorubicin liposome) [29] with 6.81 mg/mL of DSPC and 65% DSPC in the membrane component, respectively, were used. Chol is an amphiphilic substance, which can be used as a medicinal excipient, including liposome membrane materials, especially in liposome preparations. (2,3-Dioleyl-propyl)-trimethylamine (DOTAP) is a commonly used cationic lipid.

As shown in Table 1, we investigated the formulation process of the preparation by using dynamic light scanning (DLS) to evaluate the particle size, potential, polydispersity index (PDI), and particle size stability within three days. In the follow-up investigation, the optimal value obtained in the previous step was used to formulate the optimal prescription.

### 2.3. Preparations of Protein Corona Liposomes

CLs were prepared by a double-emulsion method as follows: the membrane materials, including 4 mg of DSPC, 1 mg of Chol, and 1 mg of DOTAP, were dissolved in 1 mL of dichloromethane (DCM); a total of 200 μL of phosphate-buffered saline (PBS) containing 0.6 mg of liraglutide was added and sonicated at 80 W for 1 min (ultrasonic 60 times, 1 s bursts, 1 s intervals). Colostrum was added to a 4 mL polyvinyl alcohol (PVA) solution (1% *w*/*v*). Double milk was formed at 250 W ultrasound for 1 min. The organic solvent was removed by stirring overnight to obtain the CL suspension.

The CL suspension was mixed with 10 mg/mL of BSA (dissolved in 0.9% NaCl solution) in a volume ratio of 1:1. The incubation was carried out at 37 °C for 2 h. This was followed by centrifugation at 15,000 rpm for 20 min. The precipitate was resuspended using PBS, washed by centrifugation, and repeated three times. The PcCL suspension was obtained by resuspending in 0.9% NaCl solution.

Following the above method, the Pc-AT-CL suspension was obtained by replacing Chol with Chol-PEG-AT-1002. The preparation process is shown in Figure 1.

### 2.4. Characterization of Protein Corona Liposomes

#### 2.4.1. Characterization of Liposome Nanoparticles

The particle sizes and zeta potentials of the nanoparticles were characterized by DLS. Free liraglutide was separated by ultrafiltration, and the concentrated liposome solution in the upper chamber was demulsified. The absorbance of the liposome after demulsification was measured by using an ultraviolet–visible (UV-VIS) spectroscopy (UV-2450). For PcCLs/Pc-AT-CLs, trypsin was used to degrade and shed the protein canopy, and methanol was used for demulsification. The amount of protein coating was measured using the BCA kit. The encapsulation efficiency (EE%) and the drug-loading capacity (LC%) of liraglutide were calculated using the following equations:EE% = (Demulsification liposome liraglutide content)/(Total amount of liraglutide) × 100%(1)
LC% = (Demulsification liposome liraglutide content)/(Total amount of nanocarriers) × 100%(2)

#### 2.4.2. Study on Co-Localization

To investigate whether the protein coating had coated the surface of CLs/AT-CLs well, we first performed fluorescence microscopy to observe the relative positional relationship between protein coating and CLs/AT-CLs. 1,1′-dioctadecyl-3,3,3′,3′-tetramethylindodicarbocyanine,4-chlorobenzenesulfonate salt (Did) was used to label CLs and AT-CLs, and FITC-BSA fluorescence-labeled protein corona was added to observe the co-localization relationship between the Did-AT-CLs and FITC-BSA by using fluorescence microscopy analysis.

#### 2.4.3. Preparation Characterization by Scanning Electron Microscopy (SEM) and Transmission Electron Microscopy (TEM)

The liposomes were freeze-dried into powder form, and an appropriate amount of the liposome samples were fixed on a plate using conductive glue. Gold was sprayed under a vacuum, and the morphology of the preparation was observed by using SEM.

The morphology of the liposomes was studied by using TEM following a standard procedure. Briefly, a drop of the liposome dispersion was placed on a carbon support film and allowed to adsorb. The surplus was removed using blotting paper. A drop of 1% phosphotungstic acid was added, and the liposomes were stained for 60 s. The stained liposomes were allowed to dry under ambient conditions, and TEM analysis was performed (JEM-1230; JEOL, Tokyo, Japan).

### 2.5. In Vitro Release and Evaluation of Peptide Stability

Drug release performances were studied in a simulated gastric fluid (SGF) (pH = 1.2) and a simulated intestinal fluid (SIF) (pH = 6.8). The same concentration of different liposome nano-solutions was placed in dialysis bags (100 kD, Spectrum) at 37 °C. It was placed in the SGF (10 mL) for the first 2 h and then transferred to the SIF (10 mL). At specified time points (0, 1, 2, 3, 4, 6, 8, and 10 h), 1.0 mL of the solution was removed from the release medium and replaced with an equal volume of the fresh release medium. The samples were analyzed using high-performance liquid chromatography (HPLC), and the cumulative release profile was determined.

The liraglutide solutions or liposomal nanoparticles were put into the dialysis medium and incubated overnight at 37 °C in a shaker at 100 r/min. The dialysate was taken out, concentrated to the same concentration, and subjected to circular dichroism (CD). A 400 μL sample was placed in a micro quartz dish, and the spectrum was recorded on a CD spectrophotometer with a scanning speed of 1 nm/s (190–260 nm). The sample temperature was set at 20 °C. The sample background was recorded and subtracted from the values for different samples. After the baseline subtraction, a 6-point curve smoothing was performed on the spectra to plot the CD curve.

### 2.6. Mucus Co-interaction Measurement

The binding rate of mucin to different NPs was determined as follows: first, disodium hydrogen phosphate and potassium dihydrogen phosphate were used to configure the buffer solution at pH = 6.5, and an appropriate amount of mucin was weighed to prepare the mucin solutions with concentrations of 0.1%, 0.3%, and 0.5% *w*/*v*. The Did-labeled CLs, Pc-CLs, AT-CLs, and Pc-AT-CLs were added to different concentrations of mucin solution, and three groups were parallelly established. The mixture was transferred to 37 ℃ and incubated for 30 min; it was then centrifuged at 1500 rpm for 10 min, and the supernatant was collected. Next, the aggregation of nanoparticles and mucins was studied by measuring the fluorescence intensity of the supernatant.

### 2.7. Transmucosal Transport

In situ absorption studies in rats were performed to investigate the influences of the mucus barrier on the internalization of nanoparticles in the small intestines. The SD rats were forced to fast overnight before the experiments and given free access to water. The rats were anesthetized by an intraperitoneal injection of chloral hydrate (4 mg/kg), and a midline laparotomy was performed to expose the jejunum. A 4–5 cm intestinal loop was created and ligated at both ends. The mucus layer was removed at pre-treatment or post-treatment with N-acetyl-L-cysteine (NAC). Subsequently, the pre-treated, post-treated, and un-treated intestinal tissues were mixed with 0.5 mL of the Did-labeled CLs, AT-CLs, PcCLs, and Pc-AT-CLs (3 μg/mL) for 2 h. The intestinal tissues were homogenized on a high-speed shear machine, and each sample group was quantified.

### 2.8. Biosafety Evaluation

Healthy rats were administered the drug once daily (540 μg/kg) for one week to investigate the long-term safety of drug administration. The control group did not receive any treatment. All animals were fed with normal feed and given free access to drinking water. After one week, the rats in each group were sacrificed, and their hearts, livers, spleens, lungs, and kidneys were excised after perfusion, fixed in 4% paraformaldehyde, paraffin-embedded, and stained with hematoxylin and eosin (HE). The histological status of the different tissues was observed using a light microscopy.

### 2.9. In Vivo Distribution

Twelve mice, weighing 25–30 g (whole body hair removal), were randomized into two groups of six mice each, and 1,1-dioctadecyl-3,3,3,3-tetramethylindotricarbocyanine iodide (Dir) was used label to PcCLs and Pc-AT-CLs were injected via the intra-jejunum route. The mice were dissected at 2, 4, 8, 12, 14, and 24 h after injection, and the gastrointestinal tract, heart, liver, spleen, lung, and kidney were imaged on a real-time imager (IVIS Spectrum Imaging System S091, PerkinElmer, Waltham, MA, USA). Two mice were randomly selected and anesthetized intraperitoneally. The distribution of the nanoparticles in vivo was observed at 2, 4, 8, 12, and 24 h.

### 2.10. Statistical Analysis

All data are expressed as mean ± standard deviation (SD). The statistical analysis was performed using the IBM SPSS Statistics software (version 26). One-way analysis of variance (ANOVA) was used to compare and analyze the differences among three or more groups. A *p* < 0.05 was considered statistically significant.

## 3. Results and Discussion

### 3.1. Characterization of Protein Corona Liposomes

#### 3.1.1. Process and Prescription Optimization

The application of liposomes containing a protein canopy in the field of proteins and peptides has gained traction for the use of these carriers to deliver liraglutide orally. These liposomes have good mucus penetration and intestinal cell uptake [27]. The size and surface charge of nanoparticles are known to severely affect drug delivery efficiency in vitro and in vivo [30]. To penetrate the mucus barrier and intestinal epithelial barrier and to exert a good hypoglycemic effect, there must be appropriate particle size, zeta potential, and better encapsulation efficiency.

When investigating the prescription process of CLs, small particle size, obvious positive charge, and good stability need to be considered. We first examined the mass ratio of DSPC to Chol. As shown in Figure 2, DSPC: Chol at a mass ratio of 4:1 has a particle size (Figure 2a) of 108.1 nm and a potential (Figure 2b) of 29.05 mV, showing the best particle size stability (Figure 2c) over three days and the lowest volatility. When examining the mass ratio of DOTAP to Chol, the prescription was screened based on a 4:1 mass ratio of DSPC:Chol. The three evaluation criteria of particle size (Figure 2d), potential (Figure 2e), and stability (Figure 2f) were combined, and the observations are as follows: the particle size is too large for a 3:1 ratio and the stability is poor for a 2:1 ratio. Thus, a 1:1 mass ratio was selected. Based on the drug-to-lipid ratio examination, the particle size (Figure 2g) is 128.9 nm, the potential (Figure 2h) is 28.56 mV, and the stability (Figure 2i) is the best at a ratio of 1:10. In the ultrasonic process of colostrum, due to its small volume and excessive power, it is easy to cause liquid splash. Therefore, in the colostrum process, smaller power was used to minimize the loss of excipients and drugs during the preparation. Therefore, we mainly studied the ultrasonic power of double milk. At a constant power for the colostrum, the particle size decreases with an increase in the ultrasonic power of the compound emulsion. Simultaneously, as shown in Table 2, EE% is the highest when the power is 250 W. Through the prescription and process investigation, the particle size was reduced to nearly 200 nm, the potential was close to neutral, and the best EE% was selected. In summary, the prescription process of CLs is DSPC:Chol:DOTAP = 4:1:1, drug:lipid = 1:10, a colostrum ultrasound power of 80 W, and a compound milk ultrasound power of 250 W.

By using the CLs prepared above, the prescribed dosage of the protein canopy was examined. Nanoparticles with small particle sizes, near electric neutrality, and better stability are required for penetrating the mucus layer. As shown in Figure 3, at a BSA concentration of 10 mg/mL and a volume ratio of 1:1 with CLs, the particle size is approximately 200 nm, the potential is close to electroneutrality, and the formulation’s stability is the best.

The physical and chemical properties of the AT-CLs were verified by replacing all Chol with Chol-PEG-AT 1002; the results in Table 3 show no difference in particle size potential, encapsulation rate, and drug loading with CLs.

#### 3.1.2. Characterization of Liposome Nanoparticles

In this study, PcCLs/Pc-AT-CLs were successfully prepared and used for the oral treatment of diabetes. Due to the electrostatic interaction, BSA and CLs/AT-CLs formed a core–shell structure of the nanoparticles. Liraglutide was encapsulated during liposome formation, and BSA was introduced to form a protein coating on the surface of CLs/AT-CLs. This protected the peptides that penetrated the mucosal and intestinal epithelial cells.

As shown in Table 3, the particle size of the Pc-AT-CLs is 202.9 ± 12.4 nm, and the potential is closer to the neutral charge (1.76 ± 4.87 mV), which may be due to the incompletely uniformity of the BSA-coating layer, and the zeta potential measurement is sensitive. However, the slight change in the particle size between the Pc-AT-CLs (202.9 ± 12.4) and PcCLs (209.0 ± 9.6), as well as the AT-CLs (119.6 ± 5.6) and CLs (127 ± 10.0), may be due to the enhanced interaction force of the cell-penetrating peptide AT-1002 with CLs, resulting in a slight decrease in particle size. The second goal of this preparation was to achieve high loading; the EE% in CLs is 84.63 ± 5.05%, and the drug-loading efficiency is 8.08 ± 0.48%. However, during protein coating, the drug-loading efficiency of the protein corona liposomes is 2.08 ± 0.30% due to the loss of the CL solution and the addition of protein coating.

#### 3.1.3. Co-localization Relationship

The relative positional relationship between the BSA and the inner core liposomes was investigated by using fluorescence microscopy analysis. The inner core liposomes were tracked by wrapping the red fluorescent dye, Did, inside the liposomes by drug-loading, and the protein coating was tracked by labeling BSA with the green fluorescent dye, FITC. As shown in Figure 4a, the FITC-BSA and Did-AT-CLs/Did-CLs show green and red fluorescence, respectively. Using Image J analysis, the yellow fluorescence shows their co-localization. Thus, the BSA and inner core liposomes are in the same position.

#### 3.1.4. Preparation Characterization by SEM and TEM

Since the protein crown liposome is a nano-preparation, SEM and TEM were used to determine its morphology. As shown in Figure 4b, the lyophilized AT-CLs and Pc-AT-CLs are observed by the SEM, and both are spherical. The TEM results show a distinct protein coating on the surface of the Pc-AT-CLs compared to the AT-CLs (Figure 4c), confirming successful protein crown coating.

### 3.2. In Vitro Release and Evaluation of Peptide Stability

The liraglutide release profiles from different formulations were tested by an in vitro assay. The pH was set to 1.2 and 6.8 to simulate the acidic environment of gastric and intestinal fluids in vivo, respectively [31]. As shown in Figure 5a, the amount of liraglutide released from CLs/AT-CLs is approximately 80% at 8 h, unlike the liraglutide released from the PcCLs/Pc-AT-CLs, which is significantly slower at approximately 55%. The same method was used to verify the release of pure liraglutide. The release rate of pure liraglutide was linear and rapid in the first 3 h, gradually reaching an equilibrium.

Liraglutide may be sensitive to interfacial pressure when subjected to ultrasound, so it is important to ensure the stability of liraglutide in the final product. To characterize the secondary structure of liraglutide, it was determined in the far ultraviolet region (190~260 nm) using the CD experiments (Figure 5b). The original liraglutide has two negative absorption peaks at 209 nm and 220 nm. The AT-CLs/CLs have two negative absorption peaks at 208 nm and 220 nm with a slight change; the characteristic peaks of the PcCLs/Pc-AT-CLs are unchanged. This indicates that the secondary structure of liraglutide is preserved, and the ultrasonic pressure did not cause its instability.

### 3.3. Mucus Co-Interaction

Nanoparticles with a near-neutral or negative charge and a smaller absolute value can effectively reduce the electrostatic interactions with mucus [32,33]. The lumen pH of the proximal small intestine (duodenum and jejunum) is about 6.0–7.0 [34,35,36]. Herein, we chose pH = 6.5 to model the pH in the intestinal mucus environment. To detect the effect of the protein corona modification on mucus adsorption, various concentrations of mucin solutions were prepared to evaluate the aggregation of different nanoparticles in the mucin solutions. As shown in Figure 6a, the aggregation of the PcCLs and Pc-AT-CLs decreases significantly, indicating that adsorption and aggregation occur between mucins and the CLs/AT-CLs, while the capture ability of the PcCLs/Pc-AT-CLs weakens relatively. The coating of the protein cap reverses the surface charge of the original CLs, and the nanoparticles close to electrical neutrality are more beneficial in penetrating the mucus.

### 3.4. Transmucosal Transport

To further investigate the obstructive influence of mucus on the absorption of the PcCLs and Pc-AT-CLs, in situ absorption studies were undertaken. Specifically, the mucus layer adherent to the surface of the intestine epithelium was removed at pre-treatment or post-treatment with 0.2% NAC, because NAC is a widely used mucolytic agent in clinical setting to break hydrogen and disulfide bonds of mucus [37,38]. The internalized amounts of Did are shown in Figure 6b. In the presence of mucus, the intestinal absorption of the Pc-AT-CLs is always higher compared to the AT-CLs, indicating that hydrophilic and electrically neutral preparations are more beneficial properties for mucus penetration. In the case of NAC pre-treatment, compared to the Pc-AT-CLs, the amount of intestinal internalization of the AT-CLs increases significantly, confirming that positive and hydrophobic preparations are more conducive to the internalization and absorption by intestinal epithelial cells. Simultaneously, compared to the CLs, the AT-CLs are more easily absorbed in the intestine, proving that AT-1002 plays an important role in the absorption of nanoparticles. In the case of NAC post-treatment, the uptake of the CLs and AT-CLs reduces significantly compared to the untreated samples, indicating that these are easily trapped by the mucus layer. However, the mucus layer has little effect on their absorption of PcCLs and Pc-AT-CLs, proving that the protein coating facilitates easy penetration to the mucus layer, ultimately increasing the efficiency of intestinal internalization.

### 3.5. Biosafety Evaluation

Biosafety assessment is a prerequisite for evaluating utility in future biological applications [39]. After continuous administration for a week, the viscera were dissected and stained with H&E, and the histopathological changes were observed. The microscopic examination results of the H&E staining are shown in Figure 7. There are no obvious pathological changes in the heart, liver, spleen, lung, or kidney. In general, compared to the control group, no signs of inflammatory reactions were observed in the experimental group. Thus, the oral administration of the nanoparticles prepared in this study had no obvious toxicity in the experimental animals. The PcCLs and Pc-AT-CLs are safe and effective with good application prospects.

### 3.6. In Vivo Distribution

The small animal in vivo imaging system was used to visualize the distribution of the nanoparticles in the animal models, and the in vivo distribution of the PcCLs and Pc-AT-CLs was mainly evaluated based on the luminescence intensity. As shown in Figure 8a, after administration, the fluorescence intensity of the PcCLs and Pc-AT-CLs decreases with the extension of time, and at 24 h after administration, the fluorescence intensity of both groups is weak, indicating that the drug has been digested and degraded or excreted in vitro. As shown in Figure 8b, the fluorescence intensities from other major organs (heart, liver, spleen, lung, and kidney) increase. The fluorescence intensity attenuation in the PcCL group is greater than that of the Pc-AT-CL group 2–8 h after administration, indicating that the latter has a longer absorption time. This was also verified by comparing the renal fluorescence intensity; the renal fluorescence intensity of the PcCL group is higher than that of the Pc-AT-CL group.

## 4. Conclusions

According to the properties of the mucus and intestinal epithelial barriers during oral delivery, a protein corona liposome with neutral hydrophilic properties and a core–shell structure was designed based on the characteristics of BSA and the TJ regulatory peptide, AT-1002. After penetrating the mucus layer, the outer shell gradually peeled off and exposed the core CLs that crossed the intestinal epithelial cell layer efficiently. First, liraglutide was effectively encapsulated in the CLs by the emulsion solvent evaporation method. By optimizing the preparation process, the drug EE% was improved to the best possible extent. Under the optimum conditions, the EE% of the liposomes was 84.63 ± 5.05%, and the average particle size was 202.9 ± 12.4 nm. The in vitro release of liposomes and the stability of liraglutide after release were investigated in vitro. Liraglutide showed good structural stability. The intestinal absorption potential, drug distribution, and biological safety of the adaptive protein crown liposomes were studied in vivo. The results showed that the protein corona liposomes had a good intestinal internalization effect, longer intestinal absorption time, and good biological safety. The adaptive protein corona liposomes have potential application for the oral administration of proteins and peptides.

## Figures and Tables

**Figure 1 nanomaterials-13-00540-f001:**
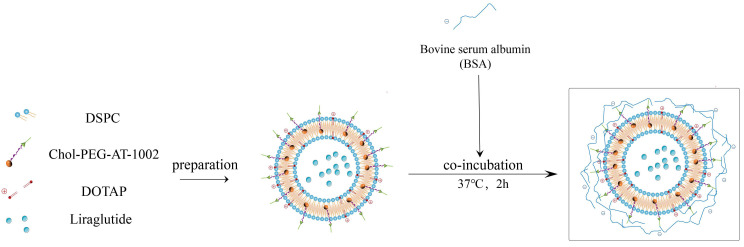
Schematic diagram of Pc-AT-CL preparation.

**Figure 2 nanomaterials-13-00540-f002:**
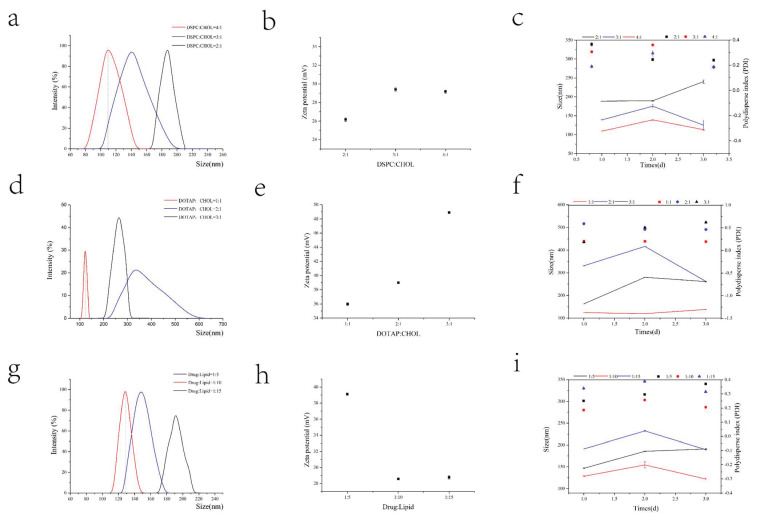
Prescription screening of cationic liposomes. The ratio of DSPC to Chol was changed while keeping other factors fixed. The particle size (**a**), potential (**b**), and particle size and PDI stability (**c**) were investigated. The optimal ratio of DSPC to Chol was adopted, and other factors were unchanged, while the ratio of DOTAP to Chol was changed. The particle size (**d**), potential (**e**), and particle size and PDI stability (**f**) were investigated. The optimal ratios of DSPC to Chol and of DOTAP to Chol were adopted, and other factors were unchanged, while the ratio of drug to lipid was changed. The particle size (**g**), potential (**h**), and particle size and PDI stability (**i**) were investigated.

**Figure 3 nanomaterials-13-00540-f003:**
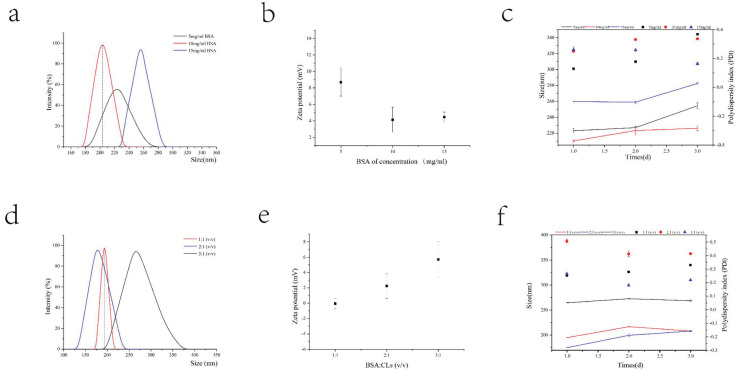
Prescription screening for protein canopy. The concentration of BSA was changed while keeping other factors fixed. The particle size (**a**), potential (**b**), and particle size and PDI stability (**c**) were investigated. The optimal concentration of BSA was adopted, and while other factors were unchanged, the volume ratio of protein solution to cationic liposome solution was changed. The particle size (**d**), potential (**e**), and particle size and PDI stability (**f**) were investigated.

**Figure 4 nanomaterials-13-00540-f004:**
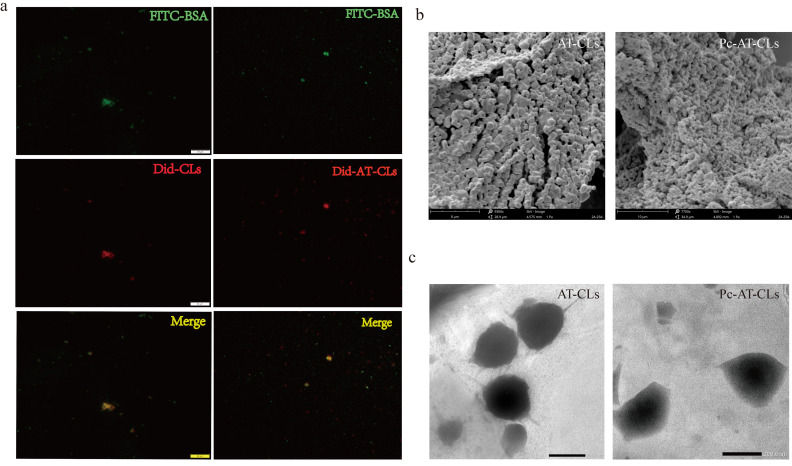
Preparation characterization of PcCL and Pc-AT-CL nanoparticles. (**a**) Visualization of the double-labeled Pc-AT-CLs by fluorescence microscopy. (**b**) SEM images of the AT-CLs and Pc-AT-CLs. (**c**) TEM images of the AT-CLs and Pc-AT-CLs.

**Figure 5 nanomaterials-13-00540-f005:**
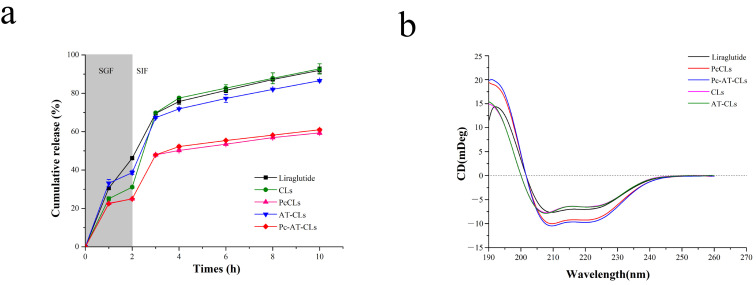
In vitro liraglutide release profiles and stability studies. (**a**) Liraglutide release profiles of the CLs, PcCLs, AT-CLs, and Pc-AT-CLs in SGF (2 h) and SIF (8 h). (**b**) Circular dichroism spectra of the liraglutide released from the CLs, PcCLs, AT-CLs, and Pc-AT-CLs (data are presented as mean ± SD, n = 3).

**Figure 6 nanomaterials-13-00540-f006:**
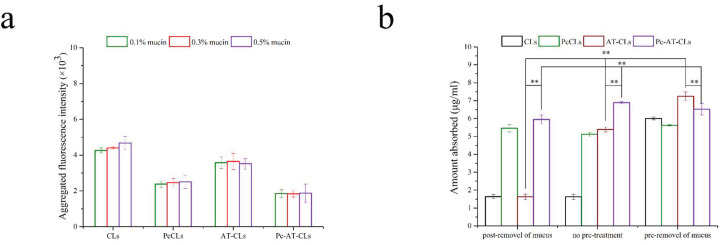
Mucus aggregation and transmucosal transport. (**a**) The binding rate of CLs, PcCLs, AT-CLs, and Pc-AT-CLs with different concentrations of mucus (data are presented as mean ± SD, n = 3). (**b**) Transmucosal transport of CLs, PcCLs, AT-CLs, and Pc-AT-CLs with or without a pretreatment or post-treatment process to remove mucus (data are presented as mean ± SD, n = 3, ** *p* < 0.05).

**Figure 7 nanomaterials-13-00540-f007:**
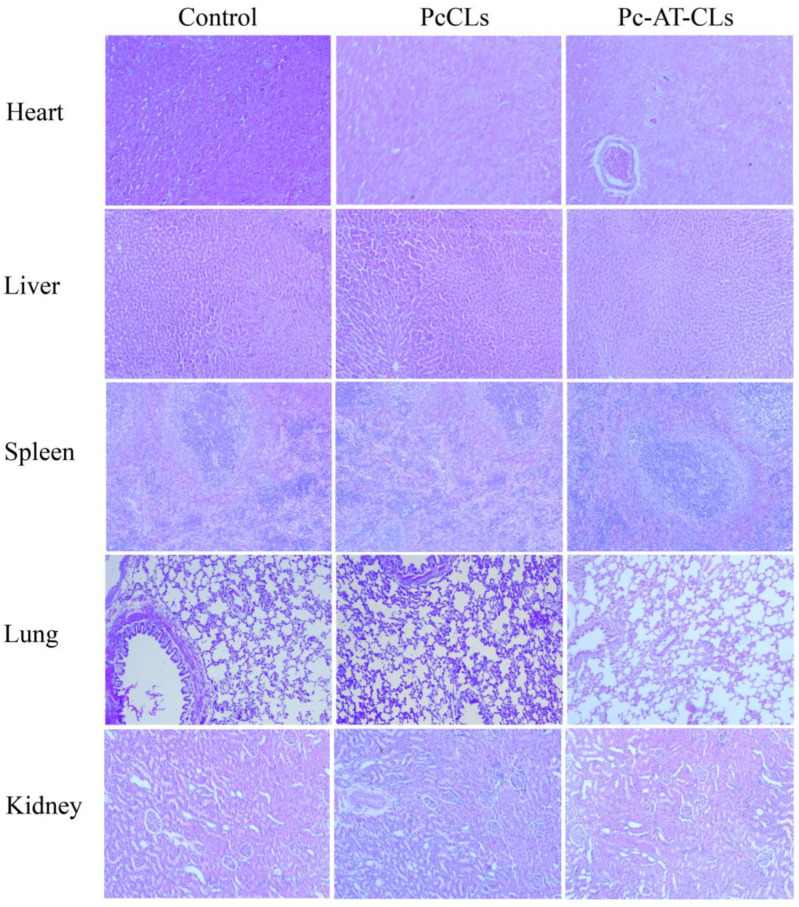
Histopathological images of the heart, liver, spleen, lung, and kidney on the 7th day, obtained using an upright fluorescence microscope (scale bar = 50 μm).

**Figure 8 nanomaterials-13-00540-f008:**
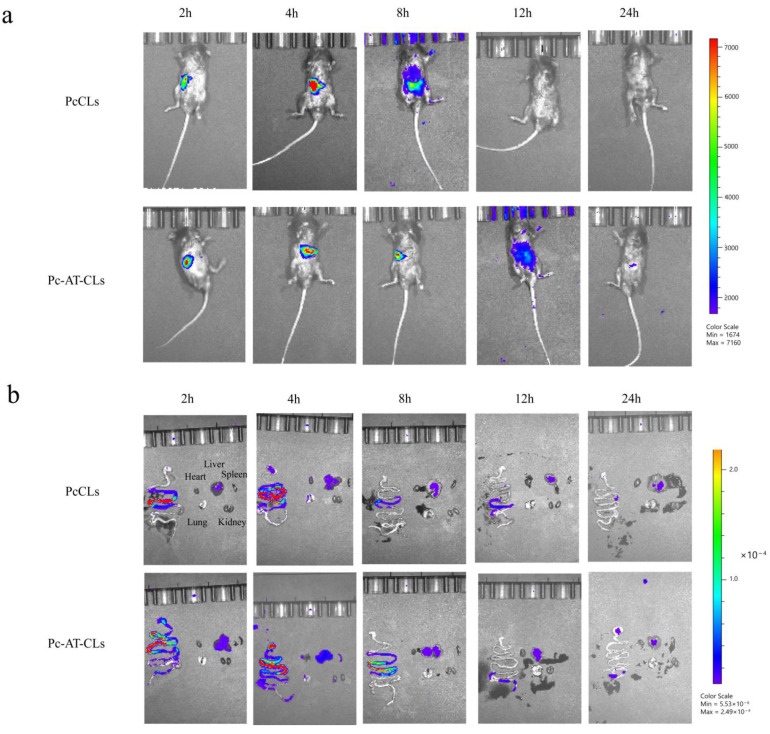
Fluorescence images of mouse organs. (**a**) After the administration of PcCLs and Pc-AT-CLs labeled with Dir, the in vivo distribution was observed at different time points. (**b**) In vivo fluorescence images of the stomach, intestine, heart, liver, spleen, lungs, and kidneys after the administration of Dir-labeled PcCLs and Pc-AT-CLs at different time points.

**Table 1 nanomaterials-13-00540-t001:** Single factor list for prescription screening.

Investigation Factors	Single Factor Ratio		
DSPC: Chol (mg/mg)	2:1	3:1	4:1
DOTAP: Chol (mg/mg)	1:1	2:1	3:1
Drug: Lipid (mg/mg)	1:5	1:10	1:15
Concentration of BSA (mg/mL)	5	10	15
BSA: CLs (*v*/*v*)	1:1	2:1	3:1
Internal water phase to oil phase (*v*/*v*)	100:1000	200:1000	300:1000

DSPC: 1,2-Dioctadecanoyl-sn-glycero-3-phophocholine; Chol: Cholesterol; DOTAP: (2,3-Dioleoyloxy-propyl)-trimethylammonium-chloride; BSA: Bovine serum albumin.

**Table 2 nanomaterials-13-00540-t002:** Investigation using ultrasonic power.

Ultrasonic Power of Colostrum (W)	Ultrasonic Power of Double Milk (W)	Zeta Potential (mV)	Size (nm)	Polydisperse Index (PDI)	Encapsulation Efficiency (EE%)
150	150	24.78	144.9	0.26	73%
80	150	21.54	140	0.24	54%
80	200	21.44	122.7	0.24	78%
80	250	27.51	120	0.29	87%

**Table 3 nanomaterials-13-00540-t003:** Size (nm), zeta potential (mV), and encapsulation efficiency (%) of the nanoparticles.

Formations	Size (nm)	Zeta (mV)	Encapsulation Efficiency (EE%)	Drug Loading (DL%)
CLs	127 ± 10.0	36.09 ± 2.7	85.85 ± 1.79	8.19 ± 0.17
AT-CLs	119.6 ± 5.6	36.14 ± 0.16	84.63 ± 5.05	8.08 ± 0.48
PcCLs	209.0 ± 9.6	0.67 ± 0.36	85.85 ± 1.79	2.92 ± 1.67
Pc-AT-CLs	202.9 ± 12.4	1.76 ± 4.87	84.63 ± 5.05	2.08 ± 0.30

## Data Availability

The data presented in this study are provided in the article.

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
