# Peer review of "Preparation, Drug Distribution, and In Vivo Evaluation of the Safety of Protein Corona Liposomes for Liraglutide Delivery"

_nanomaterials, 2023, doi:10.3390/nano13030540_

Round 1

Reviewer 1 Report

Authors have studied the application of bovine serum albumin as a shell to encapsulate liraglutide cationic liposomes, to improve its mucus permeability and prolong its absorption in the intestine.

The research was done comprehensively and properly and very good data was received. The article is very well written and reading the article is recommended for those interested and readers of Pharmaceutics.

Author Response

Thank you for your comments.

Reviewer 2 Report

This manuscript presents some quite interesting results that should be exciting to the nanomaterial world.  However, there are quite a few English corrections that must be done before publishing this manuscript.  Overall, I think this will a good manuscript once check for English flow.  My more specific comments are below:

1.       Lines 36 – 45 have little cohesion and need to be rewritten; there seems to be several different lines of thought put together that are not coherent.  Overall coherency of submission needs a thorough English review by someone with English as primary language (too many instances of incomplete sentences used).

2.       What was the pre- and post-treatments for the rats in section 2.7?  Was the ileum treated with anything pre- and post- removal?  This is not clear.

3.       Section 2.8, what were the rats administered at 540 mg/ml?

4.       Figures 2, 3, 5, 6, and 8 have axis labels that are just too blurred to follow (even at maximum magnification of pdf) and need to be of much better quality.

5.       Section 3.1.1 – have the authors tried protein-coat the liposomes after encapsulating liraglutide?

6.       Section 3.1.3 – it is not obviously clear that the authors are demonstrating that the two fluorescent probes represent the locations of BSA and core liposomes separately and that the yellow color is their overlaps which suggests that they are in the same location.  This needs to be written to make this clear.

7.       Section 3.1.4 – authors need to highlight where the protein coating is in the image.

8.       Section 3.2 – What in vivo conditions are represented at pH 1.2 and pH 6.8?

9.       Graphic abstract needs to be of higher resolution to see the words.

Author Response

  1. Lines 36 – 45 have little cohesion and need to be rewritten; there seems to be several different lines of thought put together that are not coherent.  Overall coherency of submission needs a thorough English review by someone with English as primary language (too many instances of incomplete sentences used).

Thank you for your suggestion.

Lines 36-45 have been rearranged and corrected as a whole. Simultaneously, I have also consulted a professional retouching company to revise the manuscript.

“Nanoparticles first penetrate through the mucus layer to enter the epithelial cell surface, and most nanocarrier transport relies on endocytosis (involving endosomal capture can lead to inefficient delivery) [10,11], and can also enter cells through the paracellular route [12]. The paracellular pathway is composed of tight junctions (TJs) maintained by a complex network of protein interactions. TJs are selectively permeable barriers that often represent the rate-limiting step of paracellular trafficking [13]. However, the TJ chain is a dynamic structure, and nanocarriers can temporarily reversibly open intercellular TJs by using auxiliary agents or smart formulations to penetrate cells [14]. Currently, transcellular penetration enhancers (PEs) acting on the paracellular pathway have been shown to enhance the permeability of macromolecules [15].”

  1. What was the pre- and post-treatments for the rats in section 2.7?  Was the ileum treated with anything pre- and post- removal?  This is not clear.

Thanks a lot for your comments.

(1) Pretreatment: N-acetyl-L-cysteine (NAC) treatment was performed to remove the mucus layer before incubation of the drug, with the aim of examining the intestinal epithelial absorption of the drug in the absence of the mucus layer; Post-treatment: N-acetyl-L-cysteine (NAC) treatment was performed to remove the mucus layer after incubation of the drug, with the aim of examining the intestinal epithelial absorption of the drug in the presence of the mucus layer. The retention of the formulation in the mucus was also examined in comparison with the quantification when the mucus layer was not treated.

(2) Before and after jejunostomy (After checking the manuscript, it was found that "jejunum" was wrongly written as "ileum", which has been corrected), NAC treatment was performed according to different needs according to the grouping of pretreatment group, post-treatment group and untreated group respectively, and finally both were homogenized for quantitative study.

  1. Section 2.8, what were the rats administered at 540 mg/ml?

Thanks a lot for your comments.

Examination of the manuscript revealed that the units of dose administered to rats were written incorrectly and have been corrected.

“Healthy mice were administered once daily (at a dose of 540 μg/kg) for one week to investigate the long-term safety of drug administration.”

  1. Figures 2, 3, 5, 6, and 8 have axis labels that are just too blurred to follow (even at maximum magnification of pdf) and need to be of much better quality.

Thanks a lot for your comments.

I have tried to increase the resolution of the images and have replaced the images in the manuscript with higher pixel images.

  1. Section 3.1.1 – have the authors tried protein-coat the liposomes after encapsulating liraglutide?

Thank you for your suggestion.

Protein coating has been performed after loading of liraglutide. In regard to the prescription screening for the protein canopy, the prescription used for the kernel liposomes was DSPC: Chol: DOTAP = 4:1:1, with a drug-lipid ratio of 1:10, a colostrum ultrasound power of 80 W, and a compound milk ultrasound power of 250 W. This section may not have been clearly stated in the original manuscript and has been revised in the manuscript.

  1. Section 3.1.3 – it is not obviously clear that the authors are demonstrating that the two fluorescent probes represent the locations of BSA and core liposomes separately and that the yellow color is their overlaps which suggests that they are in the same location.  This needs to be written to make this clear.

Thanks a lot for your comments.

This section has now been changed in the manuscript.

“The inner core liposomes were tracked by wrapping the red fluorescent dye Did inside the liposomes in a drug-loaded manner, and the protein coating was tracked by labeling BSA with green fluorescent FITC.”

  1. Section 3.1.4 – authors need to highlight where the protein coating is in the image.

Thanks a lot for your comments.

Scanning electron micrographs with transmission electron micrographs of the added AT-CLs were used to explain the successful encapsulation of the Pc-AT-CLs protein coating by comparison with the images of Pc-AT-CLs.

“As shown in Figure 4b, lyophilized AT-CLs and Pc-AT-CLs were observed by SEM, and both were spherical. TEM results showed a distinct protein coating on the surface of Pc-AT-CLs compared to AT-CLs (Figure 4c), confirming successful protein crown coating.”

  1. Section 3.2 – What in vivo conditions are represented at pH 1.2 and pH 6.8?

Thanks a lot for your comments.

pH 1.2: simulates the acidic environment in the stomach in vivo; pH 6.8: simulates the acidic environment in the intestine in vivo. This part has been revised in the manuscript.

“The pH was set to 1.2 and 6.8 to simulate the acidic environment of gastric and intestinal fluids in vivo.”

Reference:

Zhang, H., Gu, Z., Li, W., Guo, L., Wang, L., Guo, L., Ma, S., Han, B., Chang, J. pH-sensitive O-carboxymethyl chitosan/sodium alginate nanohydrogel for enhanced oral delivery of insulin. International journal of biological macromolecules 2022, 223, 433–445.

  1. Graphic abstract needs to be of higher resolution to see the words.

Thanks a lot for your comments.

 I redesigned and drawn the graphic abstract, applied higher resolution to the image and revised the manuscript.

Reviewer 3 Report

The submitted manuscript describes the method of preparation of protein corona cationic liposomes with osmoregulation function and their in vivo assays showing the accumulation of  nanocarriers in different organs. The physical methods proving the adequate liposomal  preparation are well decribed, as well as the biological experiments performed in this study.

However, I have some concerns about the physical characterization of the elaborated liposomes:

1) About the charge of different assemblies (Table 3), I can undersatnd that  pcCLs particles may be quasi-neutral (+0.67±0.36). But Pc-AT-CLs (zeta=+1.76±4.87) with a large error margin can be imagined as negatively charged. If the authors wish to keep the same data presentaion, some explanation on this point would be necessary.

2) As far as the size of particles (Table 3) are concerned, Pc-AT-CLs, which should have the largest dimension, has a lesser size compared to pcCLs. If the error bars are correctly determined, the only explanation might be due to a compaction effect due to the interaction of liposomes with external proteins. Is that right? Some words on this data would perhaps helpus the readers tbetter understand the phenomenon.

3) About the ECD results (Fig. 5b). Frankly, I see rather unchangeable curves showing the prepoderance of an alpha-helix fingerprint in all cases . Therefore, I cannot understand quite larg populationse  assigned to beta-strands, parallel (??),   antiparrallel (??) and random conformers. What is exactly the origin of this CD signal? peptide or protein? The bast way to compare the CD signals (otherwise the normalization of the signals would be extremely diffecult) is to show the average ellipticity per residue. But this might be hard to do.  Do the authors have other suggestions for a more accurate presentation of their ECD curves on the same graph?

Nevertheless, the total sum of the secondary structures Table 5, last row) should be 100% . Otherwise it is not a percentage. I am a bit surprized to see 100. 2%, 99.7% ..., 97.2%.  

Author Response

1) About the charge of different assemblies (Table 3), I can undersatnd that PcCLs particles may be quasi-neutral (+0.67±0.36). But Pc-AT-CLs (zeta=+1.76±4.87) with a large error margin can be imagined as negatively charged. If the authors wish to keep the same data presentaion, some explanation on this point would be necessary.

Thanks a lot for your comments.

BSA coating was performed on the same AT-CLs sample (n=3). The zeta potential of obtained samples Pc-AT-CLs is 1.76±4.87, with a large error margin, which may be due to the not completely uniformity of BSA coating layer and zeta potential measurement is sensitive.

The explanation is also added in the manuscript marked in red.

2) As far as the size of particles (Table 3) are concerned, Pc-AT-CLs, which should have the largest dimension, has a lesser size compared to PcCLs. If the error bars are correctly determined, the only explanation might be due to a compaction effect due to the interaction of liposomes with external proteins. Is that right? Some words on this data would perhaps help us the readers tbetter understand the phenomenon.

Thanks a lot for your comments.

We are very sure of the accuracy of the experimental results. For the particle size difference between PcCLs (209.0 ± 9.6) and Pc-AT-CLs (202.9 ± 12.4), maybe due to the combination of target AT-1002 and CLs, and the increase of the force between target and liposome leads to a slight decrease in particle size, which is verified by the particle size change of CLs (127 ± 10.0) and AT-CLs (119.6 ± 5.6). At the same time, this part has been supplemented in the manuscript.

3) About the ECD results (Fig. 5b). Frankly, I see rather unchangeable curves showing the prepoderance of an alpha-helix fingerprint in all cases. Therefore, I cannot understand quite larg populationse assigned to beta-strands, parallel (??), antiparrallel (??) and random conformers. What is exactly the origin of this CD signal? peptide or protein? The bast way to compare the CD signals (otherwise the normalization of the signals would be extremely diffecult) is to show the average ellipticity per residue. But this might be hard to do.  Do the authors have other suggestions for a more accurate presentation of their ECD curves on the same graph?

Nevertheless, the total sum of the secondary structures Table 5, last row) should be 100%. Otherwise, it is not a percentage. I am a bit surprized to see 100. 2%, 99.7% ..., 97.2%. 

Thanks a lot for your comments.

(1) In this study, liraglutide was selected as a model drug for liposome and BSA coating liposome delivery. To test the effect of the carrier system on liraglutide, circular dichroism is used to characterized the secondary structure of liraglutide, which is released in in vitro medium from the carrier systems. Therefore, CD curves are liraglutide characterization.

(2) CD curve is used qualitatively evaluate the secondary structure of the peptide, quantitative evaluation is too difficult.

(3) Thank you very much for your comments, we have also re-check this work and due to the instrument settings, the accuracy retains one valid digit and shows an accuracy close to 100%, where I have made changes in the manuscript.

Reviewer 4 Report

Manuscript entitled “Preparation, Drug Distribution, and in Vivo Evaluation of Safety of Protein Corona Liposomes for Liraglutide Delivery” could be interesting for the readers. However, the paper needs significant revision before publication. I have listed a few comments that need to be addressed:

1.       Introduction is short, it could be much better with more background about the work. I would advise to add some recent reviews on this topic.

2.       Add more concrete results in the abstract part.

3.       What is the novelty and importance of this work that should be clearly presented in the introduction?

4.       The schematic graphical diagram could be much better.

5.       Write the full form once when mentioning for the first instance.

6.       Improve the quality and resolution of the Figures. The figure quality is very poor. Enlarge figure 1,2,3, 5, & 6.

7.       The integration of the results from different parameters should be improved carefully.

8.       The discussion part could be much better, improve it with up-to-date citations.

9.       Conclusion could also be better.

10.    Also, carefully revise the typos and linguistic errors to make the manuscript error free.

Author Response

  1. Introduction is short, it could be much better with more background about the work. I would advise to add some recent reviews on this topic.

Thanks a lot for your comments.

I have added in the preface some recent reviews on aspects of the topic of this study, mainly concerning the latest reviews on studies of oral delivery systems. This part is marked in red in the manuscript.

“Pressing questions in the development of oral delivery systems are as follows: (1) how can drug design be guided by the information obtained from oral delivery barrier characteristics? (2) What are the stability issues of peptide drugs before they reaching the site of action? (3) Which structural features of nano formulations affect their intestinal absorption efficiency?”

“Example, researchers have constructed citric acid cross-linked shells made of carboxymethylcellulose wrapped around core nanoparticles for efficient trans-cellular transport by reversibly opening tightly connected paracellular transport [16]. Meanwhile, anthocyanins from strawberries can enhance the intestinal permeability of drugs by screening polyphenolic compounds of plant origin [17].”

“In our previous study, chitosan (CS) nanoparticles linked to AT-1002 as a core and poly-N-(2-hydroxypropyl) methacrylamide (pHPMA) as a smart escape core-shell structure overcame pH and mucus barriers and demonstrated that this delivery system significantly enhanced the oral hypoglycemic effect of liraglutide [23].”

“There are no related studies for improving the oral delivery of peptides that involve both protein-crowned liposomes combined with cellular bypass-regulating peptides. Therefore, in this study, we used BSA as a protein coating with CLs attached with an osmoregulatory peptide as the core to improve mucus permeability and enhance the absorption efficiency in the intestine.”

Reference:

[16] Li, C., Yuan, L., Zhang, X., Zhang, A., Pan, Y., Wang, Y., Qu, W., Hao, H., Algharib, S. A., Chen, D., Xie, S. Core-shell nanosystems designed for effective oral delivery of polypeptide drugs. Journal of controlled release 2022, 352, 540–555.

[17] Lamson, N. G., Fein, K. C., Gleeson, J. P., Newby, A. N., Xian, S., Cochran, K., Chaudhary, N., Melamed, J. R., Ball, R. L., Suri, K., Ahuja, V., Zhang, A., Berger, A., Kolodieznyi, D., Schmidt, B. F., Silva, G. L., Whitehead, K. A. The strawberry-derived permeation enhancer pelargonidin enables oral protein delivery. Proceedings of the National Academy of Sciences of the United States of America 2022, 119, e2207829119.

[23] Shi, Y., Liu, L., Yin, M., Zhao, Z., Liang, Y., Sun, K., Li, Y. Mucus- and pH-mediated controlled release of core-shell chitosan nanoparticles in the gastrointestinal tract for diabetes treatment. Journal of drug targeting 2023, 31, 65–73.

  1. Add more concrete results in the abstract part.

Thanks a lot for your comments.

Thanks to your suggestion, more specific results have been added to the abstract section and have been revised in the manuscript.

“Developing oral drug delivery systems is challenging, and overcoming the mucus layer and low intestinal epithelial permeability is needed. The purpose of this study was to develop an oral drug delivery system containing protein canopies and osmoregulation peptides. Cationic liposomes (CLs) linked to osmoregulatory peptides (AT-1002) were prepared by double emulsion method, and bovine serum albumin (BSA) was adsorbed on the surface of AT-CLs resulting in protein corona cationic liposome formation with AT-1002 (Pc-AT-CLs). The preparation method was determined by investigating various process parameters. The particle size, potential, and encapsulation efficiency (EE%) of Pc-AT-CLs were 202.9 ± 12.4nm, 1.76 ± 4.87mV, and 84.63 ± 5.05%, respectively. Transmission electron microscopy (TEM) imaging revealed the that nearly spherical structure of Pc-AT-CLs with a recognizable coating. Circular dichroism (CD) experiments confirmed that the complex preparation process did not affect the secondary structure of liraglutide. With the addition of protein coating and AT-1002, the mucus accumulation of Pc-AT-CLs was about 2 times lower than that of AT-CLs, and the amount of enteric metaplasia was 1.35 times higher than that of PcCLs. The intestinal absorption time of Pc-AT-CLs is longer and has good biological safety. Pc-AT-CLs showed good performance in vivo with the potential to facilitate oral liraglutide administration.”

  1. What is the novelty and importance of this work that should be clearly presented in the introduction?

Thanks a lot for your comments.

“There are few related studies for improving the oral delivery of peptides that involve both protein-crowned liposomes combined with cellular bypass-regulating peptides. Therefore, in this study, we used BSA as a protein coating with CLs attached with an osmoregulatory peptide as the core to improve mucus permeability and enhance the absorption efficiency in the intestine.”

This part has been revised in the manuscript.

  1. The schematic graphical diagram could be much better.

Thanks a lot for your comments.

I have redesigned and drawn the schematic graphical diagram, meanwhile, have made changes in the manuscript.

  1. Write the full form once when mentioning for the first instance.

Thanks a lot for your comments.

I have checked in the full text about "write in full form at the first mention" and revised it in the manuscript.

  1. Improve the quality and resolution of the Figures. The figure quality is very poor. Enlarge figure 1,2,3, 5, & 6.

Thanks a lot for your comments.

I have tried to increase the resolution of the images and have replaced the images in the manuscript with higher pixel images. This section has also been marked in red in the manuscript.

  1. The integration of the results from different parameters should be improved carefully.

Thanks a lot for your comments.

I have reorganized the " Results and Discussion" section logically and integrated and revised it.

  1. The discussion part could be much better, improve it with up-to-date citations.

Thanks a lot for your comments.

I have reorganized the discussion on logical relationships and added updated references in section 3.1.1, section 3.2, and section 3.5.

  1. Conclusion could also be better. 

Thanks a lot for your comments.

Part has been added to the conclusion, which can be made more complete.

“According to the properties of mucus barrier and intestinal epithelial barrier during oral delivery, a protein corona liposome with neutral hydrophilic and core-shell structure was designed based on the characteristics of bovine serum albumin and tight junction regulatory peptide AT-1002. After penetrating the mucus layer, the outer shell gradually peeled off and exposed the core cationic liposomes that were more conducive to cross the intestinal epithelial cell layer.”

  1. Also, carefully revise the typos and linguistic errors to make the manuscript error free.

Thanks a lot for your comments.

I checked the manuscript in its entirety, carefully corrected spelling errors and language mistakes, and consulted a professional retouching company to revise the manuscript.
